# Nematodes Consume Four Species of a Common, Wood-Decay Fungus

Abigail Ferson-Mitchell [1], Lynn Carta [2], John-Erich Haight [3] and George Newcombe [1,*]

1 College of Natural Resources, University of Idaho, Moscow, ID 83843, USA
2 Mycology and Nematology Genetic Diversity and Biology Laboratory, USDA-ARS, Beltsville, MD 20705, USA
3 Forest Mycology Research, Northern Research Station, USDA-FS, Madison, WI 53726, USA
* Correspondence: georgen@uidaho.edu; Tel.: +1-208-596-8271

**Abstract:** Since nitrogen is in short supply in wood yet relatively plentiful in the bodies of nematodes, wood-decay fungi have been thought to be nematophagous. In an earlier study, we confirmed the nematophagy of two species of wood-decay fungi (*Pleurotus ostreatus* and *P. pulmonarius*), although we also found nematode species that could turn the tables and consume *Pleurotus*. In this study, we tested interactions between nematode species and *Fomitopsis*, another genus of common wood-decay fungi. Four geographically distinct isolates, or provenances, within each of four species (i.e., the European *F. pinicola* and three North American species: *F. ochracea*, *F. schrenkii*, and *F. mounceae*) were confronted with a total of twenty nematode species (twenty-four strains) in four experiments. Nematophagy was observed much less frequently in *Fomitopsis* than in *Pleurotus*: only 31 of the 516 interactions (6%), overall, resulted in nematophagy by a *Fomitopsis* isolate, whereas with *Pleurotus*, the result was 16 of 28 (57%). In contrast, all 20 species of nematode here were capable of mycophagy and dominated interactions with all isolates of *Fomitopsis* overall. Clearly, not all wood-decay fungi are as strongly nematophagous as the *Pleurotus* species. Perhaps arboreal nematodes even tend towards mycophagy, given the limiting nitrogen in wood.

**Keywords:** nematode; mycophagy; microbial interaction; wood decomposer; wood decay; *Fomitopsis pinicola*; *C. elegans*





## 1. Introduction

Nematodes are the most abundant animals on Earth [1]. Their fossils are present in the Rhynie chert (407 Ma), along with those of ancient fungi [2]. Given their coexistence over this long period, many types of interactions between these two organismal groups have evolved. The two most basic types might be defined by consumption: fungi that consume nematodes are nematophagous, and nematodes that consume fungi are mycophagous. Among nematophagous fungi, species of *Pleurotus* stand out as well-studied models [3].

In our prior study [4] of interactions between two species of *Pleurotus* and 13 species of nematodes, 16 of 28 interactions were nematophagous, and the other 12 were mycophagous. Which organism consumed the other depended on the species identities of the interaction. Further, interactions proved to be differential. Our prior study also demonstrated that some nematode species were predators of *Pleurotus ostreatus* but prey for *Pleurotus pulmonarius*, while for a second group of nematodes, the reverse was true [4]. How complex might this interaction web be? In addition to that question, we also wondered about the rationale for nematophagy among wood-decay fungi.

Nematodes are diverse; there are 23,000 described species, but more than 1 million are predicted to exist [5,6]. Among the over 154,500 described species of fungi [7], at least 700 species are nematophagous [8]. The latter must be an underestimate, since fungi have never been systematically assayed for nematophagy. Despite the lack of testing, one guild stands out for its nematophagy: the wood-decay group that includes *Pleurotus* as

an exemplary. The rationale for probable nematophagy among wood-decay fungi is that nitrogen is in short supply in wood yet relatively plentiful in the bodies of nematodes [5,9]. In this context, it is important to remember that the nematophagy of *Pleurotus* has been demonstrated on water agar, a medium with minimal nitrogen [4]. Conversely, the same rationale would apply to nematodes in wood, that they might also overcome the limited nitrogen of their habitat by consuming fungi. Therefore, it was not entirely unexpected to find evidence of both nematophagy and mycophagy among small sets of fungi (e.g., *Pleurotus*) and nematodes, respectively, as we did in our prior study [4].

*Fomitopsis*, similar to *Pleurotus*, is a genus of common, wood-decay fungi [10]. If all wood-decay fungi are nematophagous, then species and isolates of *Fomitopsis* should be. Literature on nematophagy in *Fomitopsis* is limited; however, Balaes et al. [11] showed that one isolate of *Fomitopsis pinicola* has the nematophagous ability in nutrient-poor agar. These authors saw a higher mortality of nematodes due to *F. pinicola* on a more acidic medium, suggesting that environmental conditions influence nematophagy [5]. Further studies are needed to test whether changing *in-agaro* conditions (e.g., acidity of the medium) would affect the overall outcome of the interactions. It has also been suggested that *Fomitopsis* may paralyze nematodes and then consume them later [5,12]. However, most species of *Fomitopsis*, including some that have been recently described [13,14], have never been assayed for nematophagy. In contrast, the nematophagy of *Pleurotus* is well studied, although the chemical nature of the toxin was not elucidated until this year [15]. *Pleurotus* toxin droplets immobilize nematodes [3]. Directional hyphae then penetrate through orifices in the nematode and colonize towards the head. Armed with a toxin that acts against nematodes (one of five mechanisms employed by nematophagous fungi [12]), *Pleurotus* species seek bacteria-feeding colonies of nematodes when on nutrient-poor agar [16]. We now know that some of these bacteria-feeding nematodes are also able to consume fungi [4].

Nematophagy can vary among species of *Fomitopsis* or within species. In this study, we assayed four geographically distinct isolates, or provenances, within each of the four species (i.e., the European *F. pinicola* and three North American species: *Fomitopsis ochracea*, *Fomitopsis schrenkii*, and *Fomitopsis mounceae*) and confronted each, pair by pair, with a total of twenty-four strains of twenty nematode species (thirteen species included from the *Pleurotus* study [4]). Four experiments were conducted overall, and outcomes of interactions were assessed as they were, previously, in the *Pleurotus* study.

## 2. Materials and Methods

### 2.1. Fungal Cultures

The Center for Forest Mycology Research in Madison, Wisconsin contributed 16 cultured isolates of 4 species of *Fomitopsis*. The four species tested were *F. ochracea*, *F. schrenkii*, *F. mounceae*, and *F. pinicola*. Each species was, in turn, represented by four isolates from distinct geographic locations (Table 1). All isolates of *Fomitopsis* were maintained on potato dextrose agar (PDA) at 21 °C.

**Table 1.** Identities and source locations of four isolates for each species of *Fomitopsis* that were paired with nematode species in this study. Each isolate is referenced by the unique I.D. in this paper. Phylogenetic information is available in the study by Haight et al. [13,14]. The last column indicates interaction experiments in which each isolate was included.

| Unique I.D. | Genus | Species | Country | State | Collection Number | Experiment |
|---|---|---|---|---|---|---|
| F.m.CAN | *Fomitopsis* | *mounceae* | Canada | Alberta Province | JEH-82 | I,II |
| F.m.NH | *Fomitopsis* | *mounceae* | USA | New Hampshire | FP-125086-T | I,III,IV |
| F.m.OR | *Fomitopsis* | *mounceae* | USA | Oregon | CS-1 | I,II |
| F.m.WI | *Fomitopsis* | *mounceae* | USA | Wisconsin | JEH-147 | I,III,IV |
| F.o.AK | *Fomitopsis* | *ochracea* | USA | Alaska | LT-16 | I,III,IV |
| F.o.MI | *Fomitopsis* | *ochracea* | USA | Michigan | HHB-3331-Sp | I,II |

**Table 1.** *Cont.*

| Unique I.D. | Genus | Species | Country | State | Collection Number | Experiment |
|---|---|---|---|---|---|---|
| F.o.NH | *Fomitopsis* | *ochracea* | USA | New Hampshire | FP-125083-T | I,III,IV |
| F.o.WA | *Fomitopsis* | *ochracea* | USA | Washington | HHB-14787-T | I,II |
| F.p.EE | *Fomitopsis* | *pinicola* | Estonia | Ida-Viru | DR-EST-11 | I,III,IV |
| F.p.RU | *Fomitopsis* | *pinicola* | Russia | Moscow | TS-Fp-24 | I,III,IV |
| F.p.SE.1 | *Fomitopsis* | *pinicola* | Sweden | Östergötland | AT-Fp-1 | I,II |
| F.p.SE.2 | *Fomitopsis* | *pinicola* | Sweden | - | FCUG-2034 | I,II |
| F.s.AZ | *Fomitopsis* | *schrenkii* | USA | Arizona | RLG-10752-Sp | I,II |
| F.s.CO.1 | *Fomitopsis* | *schrenkii* | USA | Colorado | FP-105881-R | I,III,IV |
| F.s.CO.2 | *Fomitopsis* | *schrenkii* | USA | Colorado | JW-F.p.1 | I,II |
| F.s.SD | *Fomitopsis* | *schrenkii* | USA | South Dakota | JEH-150 | I,III,IV |

### 2.2. Nematode Cultures

The USDA-ARS NEA Beltsville Agricultural Research Center in Beltsville, Maryland contributed 24 strains of 20 species of bacterial-feeding nematodes representing four families (Table 2). Nematode strains were maintained on nematode growth medium (NGM) inoculated with *Escherichia coli* 'OP50' at 21 °C, and 7 mm plugs were mapped using a Zeiss Stemi dissecting scope prior to the transfers.

**Table 2.** The 24 strains of 20 nematode species paired with 16 isolates of four *Fomitopsis* species in four interaction experiments. Each strain is referenced by the abbreviated species name in this paper. Phylogenetic information can be found in the study by Carta et al. [17]. The last column indicates interaction experiments in which each strain was included.

| Unique I.D. | Family | Genus | Species | Collection Number | Experiment |
|---|---|---|---|---|---|
| 1. | Cephalobidae | *Acrobeloides* | *amurensis* | PS1146 | I,III,IV |
| 2. | Cephalobidae | *Acrobeloides* | *apiculatus* | LKC60 | I |
| 3. | Cephalobidae | *Acrobeloides* | *varius (nanus)* | LKC27 | III |
| 4. | Cephalobidae | *Acrobeloides* | *varius (nanus)* | LKC52 | I,II,III,IV |
| 5. | Cephalobidae | *Zeldia* | *punctata* | PS1192 | II,III,IV |
| 6. | Cephalobidae | *Zeldia* | *sp.* | PS1194 | II,III |
| 7. | Diplogasteridae | *Pristionchus* | *aerivorus* | LKC54 | I,II,III,IV |
| 8. | Diplogasteridae | *Pristionchus* | *entomophagus* | KR2984 | I |
| 9. | Panagrolamidae | *Panagrellus* | *redivivus* | PS1163 | I |
| 10. | Panagrolamidae | *Panagrolaimus* | *detritophagus* | LKC56 | III,IV |
| 11. | Panagrolamidae | *Panagrolaimus* | *detritophagus* | PS1162 | II,III |
| 12. | Panagrolamidae | *Panagrolaimus* | *hygrophilus* | PS1732 | II,III,IV |
| 13. | Panagrolamidae | *Panagrolaimus* | *rigidus* | LKC53 | II,III,IV |
| 14. | Panagrolamidae | *Panagrolaimus* | *sp.* | LKC39 | III,IV |
| 15. | Panagrolamidae | *Panagrolaimus* | *sp.* | LKC40 | III,IV |
| 16. | Panagrolamidae | *Panagrolaimus* | *sp.* | LKC46 | III |
| 17. | Rhabditidae | *Caenorhabditis* | *elegans* | N2 | I,II |
| 18. | Rhabditidae | *Diploscapter* | *lycostoma* | PS2017 | II |
| 19. | Rhabditidae | *Mesorhabditis* | *inarimensis* | LKC51 | I |
| 20. | Rhabditidae | *Metarhabditis* | *rainai* | LKC20 | III,IV |
| 21. | Rhabditidae | *Oscheius* | *dolichura* | LKC50 | I,II,III,IV |
| 22. | Rhabditidae | *Oscheius* | *myriophila* | DF5020 | I |
| 23. | Rhabditidae | *Oscheius* | *tipulae* | LKC57 | I,II,III,IV |
| 24. | Rhabditidae | *Poikilolaimus* | *oxycercus* | LKC64 | I,II,III,IV |

### 2.3. Fomitopsis–Nematode Interaction Experiments

In Experiment I, 5 mm fungal plugs from all 16 *Fomitopsis* isolates were transferred onto 2% water agar (WA) in 60 × 15 mm plastic petri dishes. After one week at 21 °C, the cultures were challenged with a 7 mm plug from 12 nematode species (Table 2), placed 10 mm from

the initial *Fomitopsis* inoculation. Initial observations were made using a Zeiss Stemi 2000-C dissection microscope (Carl Zeiss, Jena, Germany) after a 48 h acclimation period.

Three additional experiments were conducted in a similar manner to Experiment I, except only two isolates of each *Fomitopsis* were tested (Table 1). In Experiment II, these eight isolates were challenged after one week by six of the nematode species used in Experiment I and six additional species. Experiment III isolates were grown for two weeks, then challenged by 17 strains of nematodes consisting of 13 species. In Experiment IV, *Fomitopsis* cultures were grown for 30 days prior to the introduction of 13 strains of nematodes comprising 12 species. The nematode strains used in each experiment are listed in Table 2. Estimations of the number and age structures of the nematodes on the transferred plug were taken prior to transfer and again 48 h post transfer.

*2.4. Data Collection*

Observations were recorded, using a Zeiss Stemi dissecting scope, as fungus-dominated (F) if the fungus killed and consumed all or nearly all of the nematodes at the time of observation (Figure 1). If all or nearly all the nematodes were alive, reproducing, and feeding on the fungus at the time of observation, then those interactions were scored as nematode-dominated (N) (Figure 1). These two categories were the same as our prior study of interactions between nematodes and species of *Pleurotus*; susceptible was synonymous with fungus-dominated (F), and resistant was synonymous with nematode-dominated (N) [4].

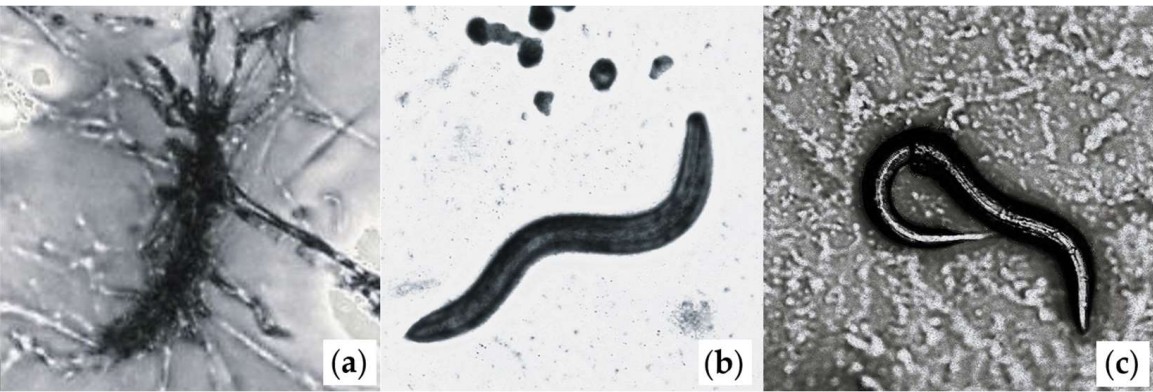

**Figure 1.** (**a**) *Fomitopsis schrenkii* 'F.s.SD' consuming a susceptible nematode species *Oscheius tipulae*, representing a nematophagous interaction. (**b**) *Acrobeloides amurensis* entirely consumed *Fomitopsis pinicola* 'F.p.EE', representing a mycophagous interaction, with evidence of nematode reproduction indicated by the presence of eggs. (**c**) *Fomitopsis mounceae* 'F.m.WI', interacting with *Metarhabditis rainai*, observed moving freely through the intact hypha.

A third category consisted of interactions that were ambiguous or uncertain (-), in which neither species appeared to be consuming the other (Figure 1). This category was further defined as the fungal mycelium remaining intact (i.e., undisturbed or uniform) and the nematodes interacting with the fungus remaining alive and moving without impairment (i.e., no trapping, paralysis, or consumption by the fungus). Further data on the status of the nematode population dynamics, in terms of age structure and reproductive status, were recorded at the time of observation. Data collection occurred 48 h, one week, and 30 days post-transfer.

**3. Results**

Nematophagy occurred at low levels in Experiment I. Of 192 interactions, only 25 (13%) interactions were dominated by one of the four isolates of *Fomitopsis* (Table 3). Conversely, mycophagy (i.e., consumption of fungi by nematodes) occurred in 148 (77%) of the interactions. The remaining 10% of interactions were ambiguous in a way that was not observed in the *Pleurotus* study [4]. *Fomitopsis*–nematode interactions developed more slowly than those

between *Pleurotus* and nematodes. Instantaneous paralysis of nematodes upon contact with *Fomitopsis* was not observed as it had been with the toxin-producing *Pleurotus* species.

**Table 3.** Outcome of Experiment I *Fomitopsis*–nematode interactions after 30 days. This experiment recorded more instances of dominance by *Fomitopsis* (i.e., 25) than any of the three subsequent experiments. The numbers in the first column correspond with nematode species in Table 2. Rows are blocked off by families in the order of: Cephalobidae, Diplogasteridae, Panagrolamidae, and Rhabditidae. N = nematodes dominated the interaction; F = fungi dominated the interaction; - = outcome in doubt or deadlocked.

| | *F. mounceae* | | | | *F. ochracea* | | | | *F. pinicola* | | | | *F. schrenkii* | | | | | | |
| --- | --- | --- | --- | --- | --- | --- | --- | --- | --- | --- | --- | --- | --- | --- | --- | --- | --- | --- | --- |
| | CAN | NH | OR | WI | AK | MI | NH | WA | EE | SE.1 | SE.2 | RU | AZ | CO.1 | CO.2 | SD | F | N | - |
| 1. | N | N | N | N | N | N | N | N | N | N | N | N | N | N | N | N | 0 | 16 | 0 |
| 2. | N | N | N | N | N | N | N | N | N | N | F | N | N | F | N | N | 2 | 14 | 0 |
| 4. | N | N | N | N | N | N | N | N | N | N | - | N | F | - | N | N | 1 | 13 | 2 |
| 7. | - | N | N | N | N | N | N | N | N | N | N | - | N | N | N | N | 0 | 14 | 2 |
| 8. | N | N | N | N | N | N | N | N | N | N | F | N | N | F | N | F | 3 | 13 | 0 |
| 9. | N | N | N | N | N | N | N | N | N | N | F | N | N | F | - | - | 2 | 12 | 2 |
| 17. | F | N | N | - | N | N | - | N | - | N | F | - | F | F | N | F | 5 | 7 | 4 |
| 19. | F | N | N | N | N | N | N | N | N | N | F | - | F | F | N | F | 5 | 10 | 1 |
| 21. | N | N | N | N | N | N | N | N | N | N | - | - | N | F | N | N | 1 | 13 | 2 |
| 22. | N | N | N | N | F | N | - | N | N | N | F | N | N | F | - | F | 4 | 10 | 2 |
| 23. | N | N | N | N | N | N | N | N | N | N | N | N | N | - | N | - | 0 | 14 | 2 |
| 24. | - | N | N | N | N | N | N | N | N | N | F | N | N | F | N | - | 2 | 12 | 2 |
| F | 2 | 0 | 0 | 0 | 1 | 0 | 0 | 0 | 0 | 0 | 7 | 0 | 3 | 8 | 0 | 4 | 25 | | |
| N | 8 | 12 | 12 | 11 | 11 | 12 | 10 | 12 | 11 | 12 | 3 | 8 | 9 | 2 | 10 | 5 | | 148 | |
| - | 2 | 0 | 0 | 1 | 0 | 0 | 2 | 0 | 1 | 0 | 2 | 4 | 0 | 2 | 2 | 3 | | | 19 |

Ten of the sixteen isolates of *Fomitopsis* were not at all nematophagous (i.e., nematodes dominated in all interactions—Table 3). Nematode dominance by mycophagy was the only interaction observed when the fungal isolate was from eastern North America (i.e., two isolates from New Hampshire and one from Michigan and Wisconsin). Only one Colorado isolate of *F. schrenkii* and one Swedish isolate of *F. pinicola* were nematophagous towards most nematodes. None of the four isolates from eastern North America were nematophagous at all, nor were three of the four European isolates of *F. pinicola*. In contrast, five of the eight isolates from western North America were nematophagous, to varying extents.

Despite low overall nematophagy in Experiment I, two isolates of *Fomitopsis* distinguished themselves, as briefly mentioned above: isolate 'F.p.SE.2' from Sweden (*F. pinicola*) and 'F.s.CO.1' from Colorado (*F. shrenkii*) were nematophagous in seven of ten and eight of ten interactions, respectively. However, in the three subsequent experiments (Appendix A) with additional nematode species and somewhat different conditions for interactions, 'F.s.CO.1' was dominated by nematodes twenty-eight times, with not a single instance of its dominance over a nematode. 'F.p.SE.2' was similarly dominated 11 times, with not a single instance of its domination over a nematode strain. The evidence from the remaining three experiments supports an overall finding for the four species of *Fomitopsis* as only weakly nematophagous; it occurred only in one of 95 interactions, zero of 135 interactions, and five of 94 interactions in Experiments II, III, and IV, respectively (Tables A1–A3). Overall, nematodes prevailed in 457 of 488 (94%) or 457 of 516 (89%) interactions when including outcomes in the total that were uncertain.

The model organism, *Caenorhabditis elegans*, which was consumed by both species of *Pleurotus*, was consumed by five isolates of *Fomitopsis* in Experiment I, with four out-

comes being uncertain. Against the remaining seven isolates of *Fomitopsis*, *C. elegans* demonstrated mycophagous consumption. In Experiment II, *C. elegans* dominated all seven isolates of *Fomitopsis* with which it was paired. Similarly, there were an additional four species of nematodes that were consumed by both species of *Pleurotus* in the prior study that dominated the interaction against some isolates of *Fomitopsis*: *Poikilolaimus oxycercus*, *Acrobeloides amurensis*, *Panagrellus redivivus*, and *Pristionchus aerivorus*.

Out of the 24 strains of nematodes represented in this study, 14 strains were repeated across multiple experiments. In total, 147 *Fomitopsis*–nematode pairings were repeated. A total of 106 pairings were repeated across two experiments, and 41 pairings were repeated across three experiments. Of these 335 interactions, 308 (92%) were nematode-dominated, 10 (3%) were *Fomitopsis*-dominated, and 17 (5.1%) interactions had uncertain outcomes. Seven *Fomitopsis* isolates were repeatedly dominated by mycophagous nematodes, regardless of the experiment (i.e., 'F.m.NH', 'F.m.OR', 'F.m.WI', 'F.o.AK', 'F.o.MI', 'F.o.WA', and 'F.p.SE.1'). In the case of fungal-dominated interactions, of the 10 incidences, all were dominated by nematodes in a previous or following experiment. Similarly, in all instances of an uncertain outcome, at least one of the experiments for that pairing had a nematode-dominated outcome (Table 4).

**Table 4.** Repeated results of Experiments I, II, III, and IV *Fomitopsis*–nematode interactions. The roman numeral below the recorded outcome is indicative of the corresponding experiment. The numbers in the first column correspond with nematode species in Table 2. Rows are blocked off by families in the order of: Cephalobidae, Diplogasteridae, Panagrolamidae, and Rhabditidae. N = nematodes dominated the interaction; F = fungi dominated the interaction; - = outcome in doubt or deadlocked.

| | *F. mounceae* | | | *F. ochracea* | | | | | *F. pinicola* | | | | | *F. schrenkii* | | | | | |
|---|---|---|---|---|---|---|---|---|---|---|---|---|---|---|---|---|---|---|---|
| | CAN | NH | OR | WI | AK | MI | NH | WA | EE | SE.1 | SE.2 | RU | AZ | CO.1 | CO.2 | SD | F | N | - |
| 1. | | N,N (I,III) | | N,N,N (I,III,IV) | N,N,N (I,III,IV) | | N,N,N (I,III,IV) | | N,N,N (I,III,IV) | | | N,N (I,III) | | N,N,N (I,III,IV) | | N,N,N (I,III,IV) | 0 | 22 | 0 |
| 4. | N,N (I,II) | N,N,N (I,III,IV) | N,N (I,II) | N,N,N (I,III,IV) | N,N,N (I,III,IV) | N,N (I,II) | N,N,N (I,III,IV) | N,N (I,II) | N,N,N (I,III,IV) | N,N (I,II) | -,N (I,II) | N,N (I,III) | F,N (I,II) | -,N,N (I,III,IV) | N,- (I,II) | N,N,N (I,III,IV) | 1 | 35 | 3 |
| 5. | | N,N (III,IV) | | N,N (III,IV) | N,N (III,IV) | | N,N (III,IV) | | N,N (III,IV) | | | N,N (III,IV) | | N,N (III,IV) | | N,N (III,IV) | 0 | 16 | 0 |
| 7. | -,N (I,II) | N,N,N (I,III,IV) | N,N (I,II) | N,N,N (I,III,IV) | N,N,N (I,III,IV) | N,N (I,II) | N,N,N (I,III,IV) | N,N (I,II) | N,N,N (I,III,IV) | N,N (I,II) | N,N (I,II) | -,N (I,III) | N,N (I,II) | N,N,N (I,III,IV) | N,N (I,II) | N,N,N (I,III,IV) | 0 | 37 | 2 |
| 10. | | N,N (III,IV) | | N,N (III,IV) | | | N,N (III,IV) | | N,N (III,IV) | | | N,N (III,IV) | | N,N (III,IV) | | N,- (III,IV) | 0 | 13 | 1 |
| 12. | | N,N (III,IV) | | N,N (III,IV) | N,N (III,IV) | | N,N (III,IV) | | N,N (III,IV) | | | N,N (III,IV) | | N,N (III,IV) | | N,N (III,IV) | 0 | 16 | 0 |
| 13. | | N,N (III,IV) | | N,N (III,IV) | N,N (III,IV) | | N,N (III,IV) | | N,N (III,IV) | | | | | N,N (III,IV) | | N,- (III,IV) | 0 | 13 | 1 |
| 14. | | N,N (III,IV) | | N,N (III,IV) | N,N (III,IV) | | N,N (III,IV) | | N,- (III,IV) | | | N,N (III,IV) | | N,- (III,IV) | | N,N (III,IV) | 0 | 14 | 2 |
| 15. | | N,N (III,IV) | | N,N (III,IV) | N,N (III,IV) | | -,N (III,IV) | | N,N (III,IV) | | | | | N,N (III,IV) | | N,N (III,IV) | 0 | 13 | 1 |
| 17. | F,N (I,II) | | N,N (I,II) | | N,N (I,II) | | | N,N (I,II) | | N,N (I,II) | F,N (I,II) | | | | N,N (I,II) | | 2 | 12 | 0 |
| 20. | | N,N (III,IV) | | N,N (III,IV) | N,N (III,IV) | | N,N (III,IV) | | N,N (III,IV) | | | | | N,N (III,IV) | | N,N (III,IV) | 0 | 14 | 0 |
| 21. | N,N (I,II) | N,N,N (I,III,IV) | N,N (I,II) | N,N,N (I,III,IV) | N,N,N (I,III,IV) | N,N (I,II) | N,N,F (I,III,IV) | N,N (I,II) | N,N,N (I,III,IV) | N,N (I,II) | -,N (I,II) | -,N,N (I,III,IV) | N,N (I,II) | F,N,N (I,III,IV) | N,N (I,II) | N,N,N (I,III,IV) | 2 | 36 | 2 |
| 23. | N,N (I,II) | N,N,N (I,III,IV) | N,N (I,II) | N,N,N (I,III,IV) | N,N,N (I,III,IV) | N,N (I,II) | N,N,F (I,III,IV) | N,N (I,II) | N,N,N (I,III,IV) | N,N (I,II) | N,N (I,II) | N,N (I,III) | N,N (I,II) | -,-,N (I,III,IV) | N,N (I,II) | -,N,F (I,III,IV) | 2 | 34 | 3 |
| 24. | -,N (I,II) | N,N (I,IV) | N,N (I,II) | N,N,N (I,III,IV) | N,N,N (I,III,IV) | N,N (I,II) | N,N,N (I,III,IV) | N,N (I,II) | N,N,N (I,III,IV) | N,N (I,II) | F,N (I,II) | N,N (I,III) | N,N (I,II) | F,N,N (I,III,IV) | N,N (I,II) | -,N,F (I,III,IV) | 3 | 33 | 2 |
| F | 1 | 0 | 0 | 0 | 0 | 0 | 2 | 0 | 0 | 0 | 2 | 0 | 1 | 2 | 0 | 2 | 10 | | |
| N | 9 | 30 | 12 | 32 | 30 | 12 | 29 | 12 | 31 | 12 | 8 | 19 | 9 | 26 | 11 | 26 | | 308 | |
| - | 2 | 0 | 0 | 0 | 0 | 0 | 1 | 0 | 1 | 0 | 2 | 2 | 0 | 4 | 1 | 4 | | | 17 |

## 4. Discussion

The interactions of wood-decay fungi affect the rate of wood decay [18] and thus the carbon cycle [19]. Since nematodes interact with wood-decay fungi, they also affect

estimates of the rate of wood decomposition [20,21]. The biotic interactions of wood decay are local-scale factors sensitive to climate [22], but many interactions among organisms of the wood-decay community are poorly studied. This has been especially true of interactions between nematodes and wood-decay fungi, in part because the presence of nematodes in deadwood had not been studied. However, a recent study of nematode diversity in deadwood (i.e., decaying logs of 13 tree species) provided a first glimpse of great nematode abundance and diversity in this habitat: 247 nematode ASVs (amplicon sequence variants) were distributed over 27 families [23]. Most families of nematodes in deadwood are thought to be comprised of bacterial and fungal feeders, similar to the species included in this study, that would encounter wood-decay fungi, such as the *Fomitopsis* species and isolates that we employed.

Interactions among basidiomycetous fungi in wood vary from exclusionary to dead-locked [24], with many intermediate outcomes. Prior to our study of *Pleurotus*–nematode interactions [4], we assumed that nematodes in the wood-decay community were either mycophagous or not mycophagous, and conversely, that fungi of that community were either nematophagous or not nematophagous. However, what we instead found was that both nematode species (i.e., thirteen species comprising four families: Cephalobidae, Diplogastridae, Panagrolaimidae, and Rhabditidae) and wood-decay fungi (i.e., two species in the white-rot genus *Pleurotus*) could be either predators or prey, depending on the specificity of the *Pleurotus*–nematode species interaction [4]. If this is generally true at the wood-decay community level, then both mycophagous nematodes and nematophagous fungi might only temporarily receive rewards of 'prey' nitrogen for their specific victories. Lasting effects would be unlikely if those rewards were rapidly turned over as the predator quickly became prey to a different, specific interactor or combatant.

This current study of 16 isolates of wood-decay *Fomitopsis* and 20 nematode species showed that some common wood-decay fungi lacked significant nematophagous ability, whereas the nematodes were all mycophagous of *Fomitopsis* to a large extent. Each of the four experiments clearly showed the dominance of nematodes over *Fomitopsis*. All twenty species, a second strain in two of the species, and three strains in one species generally dominated *Fomitopsis* isolates. Nematodes consumed the *Fomitopsis*, and they were able to reproduce (i.e., eggs and juveniles were common). This finding may be ecologically meaningful, since *Fomitopsis* species are among the most common wood-decay fungi in temperate forests [13,14]. In other words, *Fomitopsis* conks and mycelium could be important sources of food for arboreal nematodes that, again, were recently shown to be both abundant and diverse in deadwood [23].

Differential interactions were observed with two fungal isolates: *F. schrenkii* 'F.s.CO.1' and *F. pinicola* 'F.p.SE.2'. However, this finding in Experiment I (Table 3) was not confirmed in subsequent experiments for reasons that must have to do with experimental protocols and conditions that varied slightly among the four experiments. Examples of this variation could be the age of fungal cultures. In Experiments I and II, the fungal cultures were allowed only one week to establish the water agar, while Experiment III had two weeks, and Experiment IV had thirty days. In the case of Experiment IV, 30 days allowed the mycelium to be well established on the entire plate. This meant that nematodes were transferred onto older growth rather than near the hyphal edge. This also meant that the fungal culture was exposed to nutrient deprivation for a longer period of time prior to nematode introduction. This could explain the inconsistency that was observed with two repeated pairings of 'F.o.NH' and 'F.s.SD' with species in Rhabditidae, where nematophagy was observed in Experiment IV but not in earlier experiments (Table 4).

In some cases, it took one week for the fungus to consume the nematodes. In other cases, it took longer. This was not the case with *Pleurotus*, which was able to quickly paralyze the nematodes after contact. Consumption occurred within a week of introduction. The time discrepancy could be explained by the mechanism employed by the fungus. The toxin-based, nematicidal mechanism of *Pleurotus* is fast-acting [15], which explains the short window between the inception of the interaction and death. In contrast, the mechanism

of *Fomitopsis* is unclear at best [5,12], and even interaction outcomes were uncertain in prior studies. For example, Ishizaki, Nomura, and Watanabe tested for nematophagy in wood-decay fungi and found that their isolate of *F. pinicola* did not immobilize or consume nematodes [22]. Our results are in line with the latter. Even 'F.p.SE.2', a Swedish isolate of *F. pinicola*, and 'F.s.CO.1', a Coloradan isolate of *F. schrenkii*, immobilized and consumed only some species of nematodes and only under *in-agaro* conditions. Not a single isolate of *Fomitopsis* in our study consistently dominated the nematodes with which they were confronted. The species identity of nematodes is as important in determining interaction outcomes as that of the fungi [25], yet all species of nematodes in our study dominated *Fomitopsis*.

Understanding of the *Pleurotus* mechanism of nematophagy took a big step forward recently [15]. The toxin was shown to be a volatile ketone, 3-octanone, that was contained within lollipop-shaped toxocysts that emerged from *Pleurotus* hyphae. This gaseous toxin paralyzed nematodes within minutes of contact. Cells within nematode tissues (i.e., sensory neurons, muscle, and hypodermis of *C. elegans*) rapidly died following calcium influx into the mitochondria. Calcium entered cells, and organelles within cells, after 3-octanone disrupted membrane integrity. An equivalent, toxin-based mechanism is not known for *Fomitopsis*. Given the ease with which 24 strains of 20 species of nematodes consumed isolates and species of *Fomitopsis*, nematicidal toxins seem unlikely in the latter genus.

However, *Fomitopsis* was not comprehensively tested in our study. We tested only four species from the 'F. pinicola' complex that now comprises ten species [26]. In addition to the four that we tested, six new species from Asia were recently described and reported to be part of the complex [26]. Outside this complex of ten species, another thirty species also belong to *Fomitopsis*. Thus, it is possible that some as-yet-untested species of *Fomitopsis* will prove to be strongly nematophagous. It should also be said that *Fomitopsis* and *Pleurotus* do not just differ in terms of nematophagy. *Fomitopsis* causes brown rot, whereas species of *Pleurotus* cause white rot in wood. The general implications of the latter difference for fungus-feeding nematodes are not known.

What is clear from literature is that deadwood nematodes can affect the rate of wood decay, and they must, therefore, affect the release of greenhouse gases. The decay of wood blocks inoculated with a specific wood-decay fungus was slowed by the addition of a fungus-feeding nematode [27]. This was true for four species of wood-decay fungi: *Trametes pubescens*, *Ganoderma applanatum*, *Climacodon septentrionale*, and *Sphaerobolus stellatus*. However, wood decay was enhanced by adding *Aphelenchoides* sp. to a wood block inoculated with the nematophagous fungus, *Hohenbuehelia grisea*. *Hohenbuehelia* is a genus comprised of 50 nematophagous species that are closely related to *Pleurotus* [28]. Presumably, the inference is that when nematodes consumed *Trametes pubescens*, *Ganoderma applanatum*, *Climacodon septentrionale*, and *Sphaerobolus stellatus*, they slowed both the rate of decay and the rate of release of greenhouse gases. In contrast, when *Hohenbuehelia grisea* consumed the *Aphelenchoides* nematodes in deadwood, this fungus increased the rate of wood decay and the rate of release of greenhouse gases. The isolates and species of *Fomitopsis* in this study are likely to behave as *Trametes pubescens*, *Ganoderma applanatum*, *Climacodon septentrionale*, and *Sphaerobolus stellatus* did in the wood-block assay: wood-decay fungi held in check by grazing nematodes.

At times, interaction outcomes may be unclear, as a small percentage in this study were. Interactions can certainly be unclear when the mechanism is not well understood; even the chemical nature of the *Pleurotus* toxin was unknown until this year [15]. Examples of unclear implications in our own prior studies include one study in which we found that nematodes interacted with fungal endophytes of plants without affecting the plant [29]. In contrast, in a study focused on the lethal effects of *Sclerotinia sclerotiorum* on an invasive plant (*Centaurea stoebe*), a nematode (*Aphelenchoides saprophilus*) protected the plant [30]. The mechanism remains unknown, although it appears to involve interaction with *Sclerotinia*, another fungus, and the plant itself. From the wider literature, there are numerous similar examples, but one more will have to suffice for now: fungi that trap nematodes on agar that

then fail to do so when in soil [31]. While there is valuable insight that can be gained from *in-agaro* studies, the sensitivity of nematode–fungal interactions to environmental factors is fundamentally challenging. The wood-block assay, briefly discussed above, appears to be a step forward towards ecological realism for future studies of this nature [28].

## 5. Conclusions

Four species of *Fomitopsis*, in the *F. pinicola* complex, were mostly not nematophagous, at least under the conditions of these four interaction experiments. Nematophagous ability cannot, therefore, be expected in all wood-decay fungi. In contrast, all 20 nematode species and 24 strains dominated the vast majority of their interactions with the 16 isolates of *Fomitopsis* by consuming the fungus. This finding should prompt further study of the ecological significance of nematode activities in decaying wood. Nematodes, now known to be both abundant and diverse in decaying wood, may slow the rate of decay by feeding on decay fungi, even ones as common as *Fomitopsis*. By slowing the rate of wood decay, such mycophagous nematodes should be slowing carbon emissions to the atmosphere.

**Author Contributions:** Conceptualization, G.N.; methodology, G.N. and A.F.-M.; formal analysis, A.F.-M.; writing—original draft preparation, A.F.-M.; writing—review and editing, G.N., L.C., J.-E.H. and A.F.-M. All authors have read and agreed to the published version of the manuscript.

**Funding:** This research received no external funding.

**Data Availability Statement:** The raw data sets are available from the corresponding author upon reasonable request.

**Acknowledgments:** We thank FMR, Madison, WI and MNGDBI, USDA-ARS Beltsville, MD for excellent technical assistance with the cultures and the many sequences of the fungal and nematode cultures used in this study. Caitlin Mullaley and Alondra Vargas contributed to this study as Senior Project students in the Environmental Science Program at the University of Idaho.

**Conflicts of Interest:** The authors declare no conflict of interest.

## Appendix A

**Table A1.** Outcomes of Experiment II *Fomitopsis*-nematode interactions, in which nematodes dominated all but one of the interactions after one week. The numbers in the first column correspond with nematode species in Table 2. Rows are blocked off by families in the order of: Cephalobidae, Diplogasteridae, Panagrolamidae, and Rhabditidae. N = nematodes dominated the interaction; F = fungi dominated the interaction; - = outcome in doubt or deadlocked.

| | *F. mounceae* | | *F. ochracea* | | *F. pinicola* | | *F. schrenkii* | | | | |
| | CAN | OR | MI | WA | SE.1 | SE.2 | AZ | CO.2 | F | N | - |
| --- | --- | --- | --- | --- | --- | --- | --- | --- | --- | --- | --- |
| 4. | N | N | N | N | N | N | N | - | 0 | 7 | 1 |
| 5. | - | N | N | F | N | N | N | N | 1 | 6 | 1 |
| 6. | N | N | N | N | N | N | N | N | 0 | 8 | 0 |
| 7. | N | N | N | N | N | N | N | N | 0 | 8 | 0 |
| 11. | N | N | N | N | N | N | N | N | 0 | 8 | 0 |
| 12. | N | N | N | N | N | N | N | N | 0 | 8 | 0 |
| 13. | N | N | N | N | N | - | N | N | 0 | 7 | 1 |
| 17. | N | N | N | N | N | N | | N | 0 | 7 | 0 |
| 18. | N | N | N | N | N | N | N | N | 0 | 8 | 0 |
| 21. | N | N | N | N | N | N | N | N | 0 | 8 | 0 |
| 23. | N | N | N | N | N | - | - | N | 0 | 8 | 0 |
| 24. | N | N | N | N | N | N | N | N | 0 | 8 | 0 |
| F | 0 | 0 | 0 | 1 | 0 | 0 | 0 | 1 | 1 | | |
| N | 11 | 12 | 12 | 11 | 12 | 11 | 11 | 11 | | 91 | |
| - | 1 | 0 | 0 | 0 | 0 | 1 | 0 | 1 | | | 3 |

**Table A2.** Outcomes of Experiment III *Fomitopsis*-nematode interactions, in which nematodes dominated all but two of the interactions after three weeks. The numbers in the first column correspond with nematode species in Table 2. Rows are blocked off by families in the order of: Cephalobidae, Diplogasteridae, Panagrolamidae, and Rhabditidae. N = nematodes dominated the interaction; F = fungi dominated the interaction; - = outcome in doubt or deadlocked.

|  | *F. mounceae* |  | *F. ochracea* |  | *F. pinicola* |  | *F. schrenkii* |  |  |  |  |
| --- | --- | --- | --- | --- | --- | --- | --- | --- | --- | --- | --- |
|  | NH | WI | AK | NH | EE | RU | CO.1 | SD | F | N | - |
| 1. | N | N | N | N | N | N | N | N | 0 | 8 | 0 |
| 3. | N | N | N | N | N | N | N | N | 0 | 8 | 0 |
| 4. | N | N | N | N | N | N | N | N | 0 | 8 | 0 |
| 5. | N | N | N | N | N | N | N | N | 0 | 8 | 0 |
| 6. | N | N | N | N | N | N | N | N | 0 | 8 | 0 |
| 7. | N | N | N | N | N | N | N | N | 0 | 8 | 0 |
| 10. | N | N | N | N | N | N | N | N | 0 | 8 | 0 |
| 11. | N | N | N | N | N | N | N | N | 0 | 8 | 0 |
| 12. | N | N | N | N | N | N | N | N | 0 | 8 | 0 |
| 13. | N | N | N | N | N | N | N | N | 0 | 8 | 0 |
| 14. | N | N | N | N | N | N | N | N | 0 | 8 | 0 |
| 15. | N | N | N | - | N | N | N | N | 0 | 7 | 1 |
| 16. | N | N | N | N | N | N | N | N | 0 | 8 | 0 |
| 20. | N | N | N | N | N | N | N | N | 0 | 8 | 0 |
| 21. | N | N | N | N | N | N | N | N | 0 | 8 | 0 |
| 23. | N | N | N | N | N | N | - | N | 0 | 7 | 1 |
| 24. |  | N | N | N | N | N | N | N | 0 | 7 | 0 |
| F | 0 | 0 | 0 | 0 | 0 | 0 | 0 | 0 | 0 |  |  |
| N | 16 | 17 | 17 | 16 | 17 | 17 | 16 | 17 |  | 133 |  |
| - | 0 | 0 | 0 | 1 | 0 | 0 | 1 | 0 |  |  | 2 |

**Table A3.** Outcomes of Experiment IV *Fomitopsis*-nematode interactions, in which nematodes dominated all but nine of the interactions after three weeks. The numbers in the first column correspond with nematode species in Table 2. Rows are blocked off by families in the order of: Cephalobidae, Diplogasteridae, Panagrolamidae, and Rhabditidae. N = nematodes dominated the interaction; F = fungi dominated the interaction; - = outcome in doubt or deadlocked.

|  | *F. mounceae* |  | *F. ochracea* |  | *F. pinicola* |  | *F. schrenkii* |  |  |  |  |
| --- | --- | --- | --- | --- | --- | --- | --- | --- | --- | --- | --- |
|  | NH | WI | AK | NH | EE | RU | CO.1 | SD | F | N | - |
| 1. |  | N | N | N | N |  | N | N | 0 | 6 | 0 |
| 4. | N | N | N | N | N |  | N | N | 0 | 7 | 0 |
| 5. | N | N | N | N | N | N | N | N | 0 | 8 | 0 |
| 7. | N | N | N | N | N |  | N | N | 0 | 7 | 0 |
| 10. | N | N |  | N | N | N | N | - | 0 | 6 | 1 |
| 12. | N | N | N | F | N | N | N | N | 1 | 7 | 0 |
| 13. | N | N | N | N | N |  | N | - | 0 | 6 | 1 |
| 14. | N | N | N | N | - | N | - | N | 0 | 6 | 2 |
| 15. | N | N | N | N | N |  | N | N | 0 | 7 | 0 |
| 20. | N | N | N | N | N |  | N | N | 0 | 7 | 0 |
| 21. | N | N | N | F | N | N | N | N | 1 | 7 | 0 |
| 23. | N | N | N | F | N |  | N | F | 2 | 5 | 0 |
| 24. | N | N | N | N | N |  | N | F | 1 | 6 | 0 |

<div align="center">

**Table A3.** *Cont.*

</div>

| | F. mounceae | | F. ochracea | | F. pinicola | | F. schrenkii | | | | |
|---|---|---|---|---|---|---|---|---|---|---|---|
| | **NH** | **WI** | **AK** | **NH** | **EE** | **RU** | **CO.1** | **SD** | **F** | **N** | **-** |
| F | 0 | 0 | 0 | 3 | 0 | 0 | 0 | 2 | 5 | | |
| N | 12 | 13 | 12 | 10 | 12 | 5 | 12 | 9 | | 85 | |
| - | 0 | 0 | 0 | 0 | 1 | 0 | 1 | 2 | | | 4 |

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
