# Peer review of "Nematodes Consume Four Species of a Common, Wood-Decay Fungus"

_forests, doi:10.3390/f14030634_

Round 1
Reviewer 1 Report
Are all wood-decay fungi nematophagous?
The species of Pleurotus and Fomitopsis were studied to find their possible role in nematophagy. Authors have used 24 strains of 20 nematode species and 16 isolates of 4 species of Fomitopsis were used to study interactions. According to the authors not all wood decaying fungi are nematophagous. The m/s is not acceptable in the present form. The specific biochemical data for the interaction is missing. In my opinion author can either include microscopy data for interaction or biochemical activities such as extracellular chitinase of all fungal strains. Host specificity of fungi is also a major concern in nematophagy. Authors should comment on this aspect too. Authors have published similar work with Pleurotus earlier.
Author Response
In the reviewer's opinion, either microscopy or biochemical activities will provide sufficient evidence of the outcomes of interactions. We provided outcomes based on microscopy, and revised line 102 to make this explicit.
Reviewer 2 Report
Greetings and Regards
The main research question Are all wood-decay fungi nematophagous?
The introduction is well explained, but it is long and I think the extra long material should be shortened
And considering the importance of investigating different nematode species around the world, this article is suitable for publication in the Wood Science and Forest Products section.
The results of the paper, according to the authors, showed that nematophagy was observed much less often in Fomitopsis than in Pleurotus: only 31 out of 516 interactions (6%) overall, resulted in nematophagy by a Fomitopsis isolate, while with Pleurotus the result It was 16 out of 28 (57%). In contrast, all 20 nematode species here were capable of mycophagy and generally mated with all Fomitopsis isolates.
Tables 2 and 3 are scattered and need a more comprehensive explanation.
In terms of written language, it needs some changes.
Author Response
The reviewer wished to see shortening and other editorial changes. We completed the following in line with that recommendation. Line 57: deleted a word. Line 88-89: deleted some text. Replaced 'pairing' with 'interaction' throughout manuscript. Line 204: deleted some text. Final paragraph of the Discussion: deleted two sentences and some additional text.
Reviewer 3 Report
Dear authors, the work done is interesting, but the following points need to be addressed.
1. Title is vague, looks like it represents a review article.
2. you have mentioned, nitrogen is the factor for nematophagy and mycophagy, considering other nutrients (may be P or K) also play a role in this phenomenon. Please discuss this in discussion
3. Author may also refer to Jaffee (2004) wood, nematodes.....fungus A. ologospora.
Author Response
1. Title is vague. We changed it to this more specific form: Are Fomitopsis species nematophagous? 2. and 3. We added to the second-to-last paragraph of the Discussion so as to mention the A. oligospora study in which nematode trapping depended on soil versus agar, and added the Jaffee citation and reference.
Reviewer 4 Report
My comments and suggestions are included in text of ms

Author Response
In responding to other reviewers' comments, we believe we have addressed the need to edit in response to your comments.
Round 2
Reviewer 1 Report
In my opinion, authors have not revised the m/s as per the suggestion. There are no microscopy images, if any.
Author Response
Dear Reviewer:
Thank you for your comments. In response to the need to elaborate on methodology, lines 119-131 have been revised to reference microscopy and Fig. 2 A and B (i.e., photos from our prior study published in Forests). We have also added further explanation of the three categories of interaction. Repeated interactions were tabulated in Table 4 (lines 179-183). They clearly demonstrate that 'F' interactions were never successfully repeated, and more importantly that 'N' interactions dominated. The repeated interactions from Table 4 were further discussed in lines 185-195 and again in lines 237-246. In response to the need identified by the Editor to expand the word count to the recommendation of 4,000 words we expanded the text primarily in the Discussion section of the paper.”